# Mobile Edge Computing Task Offloading Strategy Based on Parking Cooperation in the Internet of Vehicles

**DOI:** 10.3390/s22134959

**Published:** 2022-06-30

**Authors:** Xianhao Shen, Zhaozhan Chang, Shaohua Niu

**Affiliations:** 1Guangxi Key Laboratory of Embedded Technology and Intelligent Information Processing College of Information Science and Engineering, Guilin University of Technology, Guilin 541006, China; 18856208448@163.com; 2School of Mechanical and Electrical Engineering, Beijing University of Technology, Beijing 100081, China; shh@bit.edu.cn

**Keywords:** Internet of vehicles, moving edge calculation, task collaborative offloading, genetic algorithm, roadside parking, mountain climbing algorithm

## Abstract

Due to the limited computing capacity of onboard devices, they can no longer meet a large number of computing requirements. Therefore, mobile edge computing (MEC) provides more computing and storage capabilities for vehicles. Inspired by a large number of roadside parking vehicles, this paper takes the roadside parking vehicles with idle computing resources as the task offloading platform and proposes a mobile edge computing task offloading strategy based on roadside parking cooperation. The resource sharing and mutual utilization among roadside vehicles, roadside units (RSU), and cloud servers (cloud servers) were established, and the collaborative offloading problem of computing tasks was transformed into a constraint problem. The hybrid genetic algorithm (HHGA) with a mountain-climbing operator was used to solve the multi-constraint problem, to reduce the delay and energy consumption of computing tasks. The simulation results show that when the number of tasks is 25, the delay and energy consumption of the HHGA algorithm is improved by 24.1% and 11.9%, respectively, compared with Tradition. When the task size is 1.0 MB, the HHGA algorithm reduces the system overhead by 7.9% compared with Tradition. Therefore, the proposed scheme can effectively reduce the total system cost during task offloading.

## 1. Introduction

In recent years, with the increase in the number of cars on the road and the progress of science and technology, cars are also moving in the direction of intelligence. The Internet of vehicles (IOV) is the application of the Internet of Things in the field of intelligent transportation and is also an important organic part of intelligent transportation systems [1]. However, intelligent vehicles on the Internet of vehicles have a large number of mobile applications, such as vehicle-mounted information communication systems and ABS. These mobile applications present some challenges for intelligent vehicles that require real-time information collection and a large number of computing resources. Due to the limited computing resources on the current intelligent vehicles, they cannot meet the requirements of low latency and high Quality of Service (QoS) [2,3,4] and may have problems in real-time and accuracy provided by mobile applications, thus increasing the potential hidden dangers in the process of driving [5,6].

In order to solve the above problems, researchers put forward the concept of mobile edge computing (MEC) [7]. Mobile edge computing provides cloud-like functionality to the end-user by equipping the edge of the network with computing and storage resources. Mobile edge computing can significantly improve the computing capacity of users, but due to the limited hardware resources of edge servers, it is still difficult to develop a better method of computing offloading [8]. When many users try to offload a task at the same time, the edge server and wireless channel take up too many resources, resulting in a long task response time [9,10]. Therefore, it is critical to design an effective offload strategy to determine which tasks to offload, which tasks to edge servers or other platforms, and how those tasks offload platforms allocate computing resources.

The composition of this article is as follows. Section 2 introduces the existing research related to offloading of mobile edge computing tasks. Section 3 introduces the system model used in this paper in detail. Section 4 introduces the algorithm used in detail. Section 5 compares its performance with existing methods. Section 6 discusses the conclusion of this paper and the future research direction.

## 2. Related Works

As a key technology in edge computing, computational offloading has been widely studied. Zhang et al. [11] proposed an offloading framework of vehicular edge computing based on a layered cloud to solve the problem of limited computing resources of MEC servers by introducing backup servers in the neighborhood. T. X. Tran et al. [12] studied joint task offloading and resource allocation in order to maximize user offloading gain. There is a need for the literature [11,12] to address the lack of computing resources for user terminals and how to optimize the resource allocation algorithm to ensure the maximum benefit of user terminals. However, there are few studies on task-specific offloading strategies.

Regarding the related research on the platform of task offloading, Mahenge et al. [13] proposed a MEC-assisted offloading scheme for energy-saving tasks, which mainly utilized the framework of cooperation between multiple MECs. This was conducted to achieve the goal of energy-saving and then propose a new wolf optimization algorithm based on the particle swarm optimization algorithm and the hybrid method to solve the optimization problems; the proposed method takes into account the effective allocation of resources, such as those for offloading subcarrier and power and bandwidth, to ensure the minimum energy consumption and latency requirements. Elham Karimi et al. [14] studied the response time required by resource allocation to ensure task offloading. However, MEC resources are limited and cannot handle the high mobility of many different applications. Therefore, the cooperation between MEC and central cloud decisions is proposed for offloading different onboard applications. Deep reinforcement learning, an appropriate computational model, is used to automatically learn the dynamics of the network state and quickly capture the best solution. Kuang et al. [15] proposed the calculation of offloading decision, cooperative selection, power allocation, and CPU cycle frequency allocation to solve the problem of minimization delay optimization while ensuring transmission power, energy consumption, and CPU cycle frequency constraints. Ni Zhang et al. [16] conceived and implemented an algorithm for calculating the combination of offloading and data cache for the MEC network cooperative system. The queuing theory was used to analyze the processing delay and transmission delay. In addition, an efficient online algorithm based on a genetic algorithm is proposed, which can customize the data cache decision according to the Spatio-temporal task popular pattern.

Regarding the research on time delay and energy consumption, L. T. Tan et al. [17] proposed a multi-time scale framework to jointly allocate cache and computing resources on the Internet of vehicles to minimize energy consumption, considering the time delay requirements of vehicle applications. Yang et al. [18] modeled the energy consumption of computational offloading from the aspects of computational task and communication and used artificial fish swarm algorithm to solve the problem of minimizing the energy consumption of computational offloading. Ning et al. [19] proposed that a cloud server collaborates with several edge servers to perform computationally intensive tasks. Qiao et al. [20] introduced the vehicle edge multi-access network to combine resource-rich vehicles with cloud servers to build a collaborative computing architecture.

C. Ma et al. [21] organized parked vehicles outside into parking clusters as virtual edge servers to assist edge servers in processing tasks. Additionally, a new task scheduling algorithm was designed to jointly determine the resource allocation of edge servers.

The offloading strategy in the literature [13] only considers offloading tasks to edge servers, and the platform of offloading tasks is single. Although collaborative offloading has been studied in the literature [14,15,16], the communication range of RSU is limited at present, leading to the failure of moving vehicles in some sections to communicate in time, and the main goal is only to reduce computational delay. Refs. [17,18] only studied the energy consumption of task unloading and Refs. [19,20] only studied the delay of task unloading. Ref. [21] proposed a new framework but mainly studied the task scheduling and task completion rate between parked vehicles and moving vehicles and took the reduction in total delay as the goal. At present, there are few studies on the time delay and energy consumption of mobile edge computing task offloading on the Internet of vehicles.

According to the investigation [22], there are always parked vehicles on both sides of urban roads, and the parking time is more than 18 h on average. Inspired by this, roadside parked vehicles have idle computing resources, which can be used as a platform for offloading mobile edge computing tasks.

The contributions of this paper are as follows:A moving edge computing framework based on roadside parking cooperation is proposed. In the case of no RSU or insufficient vehicle local computing resources, roadside parking was added as an offloading platform;After the global optimal solution was generated by the crossover and variation in the traditional genetic algorithm, a mountain-climbing algorithm was added to search for the local optimal solution, which improves the convergence speed and reduces the system overhead;In order to evaluate the proposed task offloading scheme based on a hybrid genetic algorithm, it was compared and analyzed with Local, ATM, Random, and Tradition task offloading methods in aspects of system overhead, delay, and energy consumption;Finally, we evaluated our method in detail from two aspects: task number and task size. Our scheme is superior to the other four offloading schemes in system overhead, delay, and energy consumption. In other words, our method produces less system cost for the same task guarantee, or equivalently, it provides a higher quality of service guarantee for the same system cost.

## 3. System Model

### 3.1. Network Model

As shown in Figure 1, in the scenario of ordinary roads, there are N mobile vehicles V, M RSUs, roadside parked vehicles, and cloud servers in total. MEC servers are deployed in each RSU to provide computing resources for vehicles. Computing resources between vehicles, RSUs, and cloud servers can be shared and used with each other. Roadside parked vehicles with no computing tasks, RSUs, and cloud servers can assist vehicles that need computing. Tasks on a moving vehicle can be calculated locally, offloaded to off-road parking, migrated to an RSU, or loaded into a cloud server over a cellular network for processing.

### 3.2. Communication Model

Assume that all vehicles are traveling at a constant speed S. Each car has a computational task; define the task as $T_{i}=\left\{d_{i},b_{i},f_{i},t_{i}^{max}\right\}$, i∈N={1,2,3,⋯,N}. Here, $d_{i}$ represents the size of the input data used in the calculation, and $b_{i}$ indicates the computing resources required to complete the task $T_{i}$.$f_{i}$ indicates the computing resource i of the vehicle, $t_{i}^{max}$ indicates the maximum delay constraint $T_{i}$ of the task, the computing resource of MEC is $f^{mec}$. Assuming that all vehicles have the same transmitting power, the data transmission rate between moving vehicles, roadside vehicles, and RSU is as follows:(1)$$R_{i}^{1}=B\log_{2}(1+\frac{QW^{2}}{N_{0}})$$
where B represents the channel bandwidth, Q represents the transmitting power of vehicles on the channel, W represents the channel gain of moving vehicles on the channel to RSU, and $N_{0}$ represents the white noise power.

Since tasks can be calculated locally, on MEC servers, on cloud servers, and on roadside parked vehicles, it is necessary to divide tasks and determine which platforms to offload to. This paper defines $S=\left\{s_{i}|s_{i}\in \left\{s_{i}^{c},s_{i}^{mec},s_{i}^{cloud},s_{i}^{side}\right\},i\in N\right\}$ as the offloading decision of the vehicle. $s_{i}^{c}$, $s_{i}^{mec}$, $s_{i}^{cloud}$, and $s_{i}^{side}$ indicates that the task is offloaded to the local server, MEC server, cloud server, and parked vehicle, respectively.

### 3.3. Calculation Model

#### 3.3.1. Local Computing Model

When the task of moving the vehicle is calculated locally, let $t_{i}^{c}$ represent the local execution delay of the moving vehicle. $t_{i}^{c}\;$ indicates the local processing delay, and $e_{i}^{c}$ indicates the local processing power consumption.
(2)$$t_{i}^{c}=\frac{b_{i}}{f_{i}}$$
(3)$$e_{i}^{c}=t_{i}^{c}p_{i}$$
where $p_{i}$ represents the equipment power of moving vehicle i.

#### 3.3.2. MEC Calculation Model

When a vehicle chooses to offload its tasks onto the MEC server, the delay can be divided into three parts. The first part is the transmission delay $t_{i}^{m1}$ required by the task offloaded by the moving vehicle to reach the MEC server. Since the MEC server is equipped on the RSU, this part of the time is equal to the delay of the task to reach the RSU.
(4)$$t_{i}^{m1}=\frac{d_{i}}{R_{i}^{1}}$$

The second part is the execution delay $t_{i}^{m2}$ of this task on the MEC server.
(5)$$t_{i}^{m2}=\frac{b_{i}}{f_{i}^{mec}}$$
where $f_{i}^{mec}$ represents the number of resources allocated from the MEC server to the vehicle to offload the task.

The third part is the return delay $t_{i}^{m3}$ of the task file from RSU to the moving vehicle.
(6)$$t_{i}^{m3}=\frac{\lambda d_{i}}{R_{i}^{1}}$$
where λ is the coefficient of output data quantity, representing the relationship between output data quantity and input data quantity.

The total delay $t_{i}^{mec}$ and total energy consumption $e_{i}^{mec}$ of offloading tasks from mobile vehicles to MEC servers are as follows:(7)$$t_{i}^{mec}=t_{i}^{m1}+t_{i}^{m2}+t_{i}^{m3}$$
(8)$$e_{i}^{mec}=t_{i}^{m1}p_{i}^{up}+(t_{i}^{m2}+t_{i}^{m3})p_{mec}$$
where $p_{i}^{up}$ represents the power of moving vehicles to upload tasks, and $p_{mec}$ represents the power of moving edge computing servers.

#### 3.3.3. Cloud Server Computing Model

Mobile vehicles offload their computing tasks to cloud servers thousands of miles away via fiber optics and core networks [23]. Therefore, the upload time for transferring input data from the RSU to the cloud server must be considered. In addition, although the amount of output data is much smaller than the amount of input data, the download time to send the results from the cloud server back to the RSU is not negligible. In this case, let $t_{i}^{y1}$ represent the execution time of the task in the cloud server, $t_{i}^{y2}$ represent the transmission delay of the task to RSU, $t_{i}^{y3}$ represent the return delay of the calculation result from RSU to the mobile vehicle. On the optical fiber line, the average transmission wait delay of tasks is calculated as $t_{cloud}$, $t_{i}^{cloud}$ represents the total delay of offloading tasks from the moving vehicle to the cloud server, and $e_{i}^{cloud}$ represents the total energy consumption of offloading tasks from the moving vehicle to the cloud server.
(9)$$t_{i}^{y1}=\frac{b_{i}}{f_{i}^{\mathrm{cloud}}}$$
(10)$$t_{i}^{y2}=\frac{d_{i}}{R_{i}^{1}}$$
(11)$$t_{i}^{y2}=\frac{\lambda d_{i}}{R_{i}^{1}}$$
(12)$$t_{i}^{cloud}=t_{i}^{y1}+(t_{i}^{y2}+t_{cloud})+(t_{i}^{y3}+t_{cloud})$$
(13)$$e_{i}^{cloud}=t_{i}^{y1}p_{cloud}+t_{i}^{y2}p_{i}^{up}+t_{i}^{y3}p_{RSU}$$
where $f_{i}^{cloud}$ represents computing resources provided by the cloud server for mobile vehicles, $p_{cloud}$ represents device power of the cloud server, and $p_{RSU}$ represents RSU transmitting power.

#### 3.3.4. Calculation Model of Roadside Parking

When a moving vehicle chooses to offload its task onto a roadside stop, the delay can be divided into three parts.

The first part is the task execution delay of roadside parking.
(14)$$t_{i}^{s1}=\frac{b_{i}}{f^{side}}$$
where $f^{side}$ represents the computing resources provided by roadside parking.

The second part is the transmission delay $t_{i}^{s2}$ of the task from moving vehicles to roadside parking.
(15)$$t_{i}^{s2}=\frac{d_{i}}{R_{i}^{1}}$$

The third part is the delay $t_{i}^{s3}$ of sending the result to the moving vehicle after the task is processed.
(16)$$t_{i}^{s3}=\frac{\lambda d_{i}}{R_{i}^{1}}$$

The total time delay $t_{i}^{side}$ and total energy consumption $e_{i}^{side}$ of offloading tasks from moving vehicles to roadside parking are as follows:(17)$$t_{i}^{side}=t_{i}^{s1}+t_{i}^{s2}+t_{i}^{s3}$$
(18)$$e_{i}^{side}=(t_{i}^{s1}+t_{i}^{s3})p_{side}+t_{i}^{s2}p_{i}^{up}$$
where $p_{side}$ represents the power of roadside parking equipment.

In summary, the total delay T for processing the whole task in this strategy can be given by the following formula:(19)$$T=\sum_{i=1}^{n}\left(t_{i}^{c}+t_{i}^{mec}+t_{i}^{cloud}+t_{i}^{side}\right)$$

Similarly, calculating the total energy consumption E of a task can be given by the following formula:(20)$$E=\sum_{i=1}^{n}\left(e_{i}^{c}+e_{i}^{mec}+e_{i}^{cloud}+e_{i}^{side}\right)$$

### 3.4. Problem Expression

This paper attempted to minimize the delay and energy consumption of the entire system and analyze the impact of task T and task E offloading to find a solution that makes the total cost more suitable for the real scenario. The linear combination of time delay and energy consumption provides a flexible method for system cost measurement by adaptively adjusting the linear weight factor when performing complex calculation tasks. Specifically, when the weighting factor is 0 or 1, only delay or energy consumption is considered. In this paper, the weighted sum of delay and energy consumption of the whole system was considered, and a weight factor: α∈[0,1] was introduced, so different weights can be given according to specific needs.
(21)W=αT+(1−α)E
where W represents the cost of the collaborative offloading system.

In this paper, task offloading and resource allocation were formulated as optimization problems to minimize W. The optimization problem is expressed as:(22)$$\underset{S,F}{\max}\sum_{i=1}^{n}W\;s.t.\;C1:s_{i}^{c},s_{i}^{mec},s_{i}^{cloud},s_{i}^{side}\in [0,1]\;s_{i}^{c}+s_{i}^{mec}+s_{i}^{cloud}+s_{i}^{side}=1,\forall i\in [1,n]\;C2:\max(t_{i}^{c},t_{i}^{mec},t_{i}^{cloud},t_{i}^{side})\leq t_{i}^{\max},\forall i\in [1,n]\;C3:\sum_{i=1}^{N}f_{i}^{mec}\leq f^{mec},i\in [1,n]$$
where S indicates the offloading decision of the vehicle, and F indicates the allocation of computing resources, that is, $F=\left\{f_{1}^{mec},f_{2}^{mec},\cdots f_{N}^{mec}\right\}$. C1 indicates that only one uninstallation platform can be selected for each task. C2 indicates that the time to complete the task of each vehicle shall not exceed the maximum allowable delay; C3 indicates the constraint on the total computing resources of the MEC server.

## 4. A Hybrid Algorithm Based on Hill-Climbing Algorithm and Genetic Algorithm (HHGA)

The Genetic Algorithm (GA) [24] is an Evolutionary Algorithm. In nature, individuals who can adapt to changing conditions can survive, while others cannot, and individual characteristics are written on genes stored on chromosomes. Compared with individuals with poor environmental adaptability, individuals with good environmental adaptability are more likely to survive. Hill-climbing algorithm [25] is a local search method. It is an iterative algorithm, and the hill-climbing algorithm is suitable for finding local optimal values, but it cannot guarantee to find global optimal values outside the search space, especially when there are multiple local optimal values.

However, a mountain-climbing algorithm can be used as an assistant to find local optimal when the genetic algorithm is dealing with complex problems. The mountain-climbing algorithm is regarded as an operator and placed in a genetic algorithm to increase the local search ability of the genetic algorithm and to improve the convergence and stability of the algorithm. Therefore, in this paper, a hybrid algorithm based on a hill-climbing algorithm and genetic algorithm HHGA (a hybrid algorithm based on a hill-climbing algorithm and genetic algorithm) was introduced to solve the multi-constraint problem. Figure out the best strategy.

### 4.1. Integer Coding and Initial Population

In this paper, each offloading strategy was regarded as a chromosome by integer coding, and each chromosome has N genes. The genes on the chromosome have three possible values −1, 0, 1, and 2, corresponding to local calculation, MEC server calculation, cloud server calculation, and roadside parking calculation, respectively. The encoding is shown in Figure 2.

*M* individuals were randomly generated as the initial population P(m).

### 4.2. Fitness Function

In searching for an optimal solution, a fitness function was used to evaluate a possible solution (individual). The fitness function determines the strongest individuals with high fitness values, and these strongest individuals are selected as the parents of the next generation. Let the reciprocal of Equation (21) be the fitness function. When the value of Equation (23) is higher, it means that the delay and energy consumption of this offloading scheme is smaller; that is, the offloading scheme is better.
(23)$$Fitness=\frac{1}{W}=\frac{1}{\alpha T+(1-\alpha )E}$$

### 4.3. Select Operations

The selection operation is the basic method of genetic algorithm to achieve good gene transfer. In this paper, the roulette selection method was used [26]. In this selection, the fitness of each chromosome was assessed by the fitness function described above. The steps to solve the maximization problem with this selection method are as follows:(1)The fitness value of individuals in the population is superimposed on the total fitness value of 1;(2)The fitness value of each divided by the total fitness is worth the probability $P_{i}$
of individual selection;
(24)$$p_{i}=\frac{f_{i}}{\sum_{j=1}^{M}f_{j}}$$
where *M* is population size.

(3)Calculate the cumulative probability $q_{i}$ of individuals to construct a roulette wheel;


(25)
$$q_{i}=\sum_{j=1}^{i}p_{i}$$


(4)Generate a random number within the interval of [0, 1]. If the random number is less than or equal to the cumulative probability of the individual and greater than the cumulative probability of individual 1, select the individual to enter the offspring population.

Repeat step (4) times and the obtained individuals constitute a new-generation population.

### 4.4. Cross Operations

This paper chose a method of uniform crossover [27]. For each gene of the first offspring, a number u∈[0,1] is uniformly generated in order to determine which parent it will inherit the gene from according to the following conditions.
(26)$$\{\begin{array}{c} g\left[i\right]\leftarrow pr_{1}\left[i\right],if\;u\geq h\\ \;g\left[i\right]\leftarrow pr_{2}\left[i\right],otherwise \end{array}$$
where g[i] represents the ith position of the offspring chromosome, ∈{1,2} of $Pr_{j}\left[i\right]$ is the ith position on the paternal chromosome j.  h∈[1,2] is named crossover rate, indicating the threshold selected. Normally, each gene of the first progeny is selected with probability h from either parent, and each gene of the second progeny is selected from the corresponding parent selected by the first progeny gene. The uniform crossover process is shown in Figure 3.

### 4.5. Mutation Operation

Adaptive variation is adopted in this paper [28], and the formula is as follows:(27)$$p_{m}^{\left(n+1\right)}=\eta p_{m}^{0}\sqrt{\frac{\sum_{i=1}^{m}{\left(f_{\max}^{\left(n+1\right)}-f_{i}^{\left(m\right)}\right)}^{2}}{\sum_{i=1}^{m}{\left(f_{\max}^{\left(n\right)}-f_{i}^{\left(n\right)}\right)}^{2}}}$$
where $p_{m}^{0}$ represents the first-generation variation rate, $p_{m}^{n}$ represents the variation rate of the n generation, $p_{m}^{n+1}$ represents the variation rate of the n+1  generation,$\;f_{i}^{\left(n\right)}$ represents the fitness value of the n generation individual, $f_{max}^{n+1}$ represents the highest fitness value of the n+1 generation, and η is the adjustment coefficient.

### 4.6. Climbing Operation and Termination Rules

Mountain climbing is a search algorithm with a good local optimization effect. First, add random point in the search space as the initial iteration point, then randomly generate within their neighborhood, calculating the function value; if the function value is superior to the point at which the current point, the initial points are replaced with the current point as a new initial point continue to search in the neighborhood, or continue to another point in the neighborhood randomly generated comparing with the initial point. The search process terminates until it finds a point that is better than it or fails to find a point that is better than it for several consecutive times. The mountain climbing method can quickly converge to the local optimal point when dealing with problems, but the multi-peak problem has multiple peak points, and the mountain climbing method can only find one of the local optimal points, not necessarily the global optimal point, so the global optimal point cannot be determined. Although global optimization cannot be a mountain-climbing method, the climbing method has the advantage that traditional optimization algorithm does not have, which is that the climbing method can handle non-micro unimodal functions because the climbing method by random individual optimization in the neighborhood does not need to use gradient, so the climbing method can deal with complex problems in the genetic algorithm (ga) that play a role of local optimization.

The new generation population selected after mutation is optimized by a mountain-climbing algorithm so that individuals can achieve a better local optimal.

The condition of ending the algorithm is that the value of the fitness function remains unchanged or reaches the specified number of iterations. Otherwise, the algorithm starts to search for a new population and starts a new iteration until the algorithm terminates. The process of Algorithm 1 is shown below.
**Algorithm 1:** HHGA algorithmInput: Population size, *M*              Selection probability, $P_{i}$
              Crossover probability, h
              Mutation probability, $P_{m}^{0}$
              Number of iterations, gen  Output: W1.t = 0;2.Initialize P(m);3.Repair P(m);4.Calculate $Fitness=\frac{1}{W}=\frac{1}{\alpha T+(1-\alpha )E};$5.Store best solutions of P(m) in old B(m);6.while t < gen do7.      Selection operation $p_{i}$,$q_{i}$ to P(m);8.       Crossover operation g[i] to P(m);9.       Mutation operation $P_{m}^{\left(n+1\right)}$ to P(m);10.     hill-climbing operation to P(m);11.     Store the best fitness individuals of P(m) in new B(m); *12.*      if Fitness(old B(m)) > Fitness(new B(m))then               new B(m)=o ld B(m)13.     end if*14.*     old B(m) = new B(m)15.     find the worst fitness value in P(m) and replace it with new B(m);16.     t = t + 1;17.end while

## 5. Simulation Verification and Analysis

### 5.1. Simulation Parameter Setting

The simulation scene of this paper was on the highway, which is 2100 m long and has two lanes. Each lane was 3.75 m wide. Vehicles were randomly distributed on the road, traveling at speeds of 30 to 50 kilometers per hour. By default, there were 30 vehicles on the road and the task size $d_{i}$ ranged from 0 to 2 MB, the maximum delay constraint $t_{i}^{max}$ was 5 s. This paper carried out the simulation through MATLAB. The relevant parameters in this paper were set under the constraints of the IEEE 802.11P standard and the mobile edge computing white paper, and some adjustments were made according to the simulation environment. In order to simplify the model, we consider that only the Gaussian white noise $\;N_{0}$ value was −100 dBm, and there are no other interference factors. Specific simulation experiment parameters are shown in Table 1.

### 5.2. Comparison Scheme Settings

In order to better evaluate the algorithm, the task offloading strategy based on roadside parking cooperative moving edge computing was compared with other the four task offloading strategies:Strategy 1: Moving Vehicle Local Execution Policy (Local): all tasks need to be executed only on the moving vehicle;Strategy 2: MEC Server Policy (ATM): all tasks need to be offloaded and executed on the MEC server;Strategy 3: Random Offloading Policy (Random): tasks are randomly offloaded on moving vehicles, MEC servers, roadside vehicles, and cloud servers;Strategy 4: Traditional Genetic scheme (Tradition) [29,30]: the classical genetic algorithm was used to realize task offloading processing for the overhead model established in this paper.

### 5.3. Impact of Number of Tasks on Algorithm Performance

Figure 4 shows the changes in system overhead of five offloading schemes as the number of iterations increases. As it can be seen from Figure 4, the system overhead of the HHGA algorithm is lower than that of the GA algorithm. Since the GA algorithm tends to fall into local optimum, the HHGA algorithm added to a mountain-climbing algorithm can improve this problem well. Both HHGA genetic algorithm and GA algorithm are superior to Local, ATM, and Random. Even though the system overhead of the Random algorithm is lower in rare cases, the overall overhead of the Random algorithm was much higher than that of the HHGA algorithm in this paper. Because the Random algorithm offloads tasks to the cloud through Random selection, tasks were randomly offloaded to mobile vehicles, MEC servers, roadside vehicles, and cloud servers for execution, which has certain randomness. As the number of iterations increases, the Local algorithm stays the same because all tasks are executed locally with no transport costs, but the system overhead is still high. Compared with the other four offloading schemes, the task offloading strategy proposed in this paper was based on roadside parking cooperative moving edge computing achieves the minimum system overhead.

Figure 5 shows the delay changes of five offloading schemes as the number of tasks increases. It can be seen from Figure 5 that the Local algorithm has the highest latency, which indicates that the offloading of computational load can reduce the execution time of the task. When the number of tasks is 20, the delay of the ATM algorithm suddenly increases because when the number of tasks reaches a certain value, the computing resources allocated by MEC to vehicles carrying tasks are not as much as the local resources, so the delay of ATM algorithm is higher than the Local delay. When the number of tasks is 25, the delay performance of the HHGA algorithm improves by 52.1%, 27.6%, 28.8%, and 24.1%. It can be seen that with the increase in the number of tasks, the delay of different strategies increases. Due to the limited computing capacity of terminal devices, channel band resources are limited. As the number of tasks increases, the load of computing equipment increases, and the limited wireless resources cannot cope with the increase in the number of tasks. Because the HHGA algorithm can quickly make offloading decisions and effectively optimize the system costs, the delay of the HHGA algorithm is minimal, respectively, compared with Local, ATM, Random, and Tradition. Therefore, the HHGA algorithm in this paper had the minimum delay compared with other offloading schemes.

As it can be seen from Figure 6, the energy consumption of the ATM algorithm increases rapidly as the number of tasks increases. This is because the ATM algorithm does not consider the computing resources of a cloud server, which brings greater energy consumption to the moving vehicle. Compared with ATM algorithms, Random, HHGA, and Tradition algorithms of cloud servers have obvious energy consumption reduction. As the number of tasks increases, the offloading scheme proposed in this paper has smaller energy consumption compared with other schemes.

### 5.4. Impact of Task Size on Algorithm Performance

Figure 7 shows the system overhead of different task sizes, and it can be seen that the Random algorithm has fluctuations because it can be randomly offloaded to mobile vehicles, MEC servers, roadside vehicles, and cloud servers for execution. For the other four algorithms, when the task input data increases, the system overhead increases with the task input data. When the task size is 1.0 MB, the system overhead of the HHGA algorithm is reduced by 42.4%, 41.3%, 39.3%, and 7.9%, respectively, compared with Local, ATM, Random, and Tradition. Therefore, the HHGA algorithm proposed in this paper had the minimum system overhead, and the speed of rising was relatively slow.

## 6. Conclusions

This paper proposed an offloading strategy for roadside parking coordinated moving edge computing tasks. In this scheme, roadside parking was added as the tasks offloading platform, and three tasks offloading platforms, including local, RSU, and cloud servers, were combined for task processing. An HHGA algorithm was proposed, and a mountain-climbing algorithm was introduced based on the GA algorithm to improve the problem that the GA algorithm falls into local optimal. Experimental results show that compared with other offloading schemes, the proposed scheme can effectively reduce system overhead, delay, and energy consumption based on the number of tasks or task size and was superior to the other four offloading schemes. Therefore, the task offloading strategy scheme of moving edge computing coordinated with roadside parking can reduce the total system cost during task offloading more effectively.

In the future, the density of roadside parked vehicles and the moving speed of moving vehicles will be considered, and new heuristic algorithms will continue to be explored for offloading processing of collaborative moving edge computing.

## Figures and Tables

**Figure 1 sensors-22-04959-f001:**
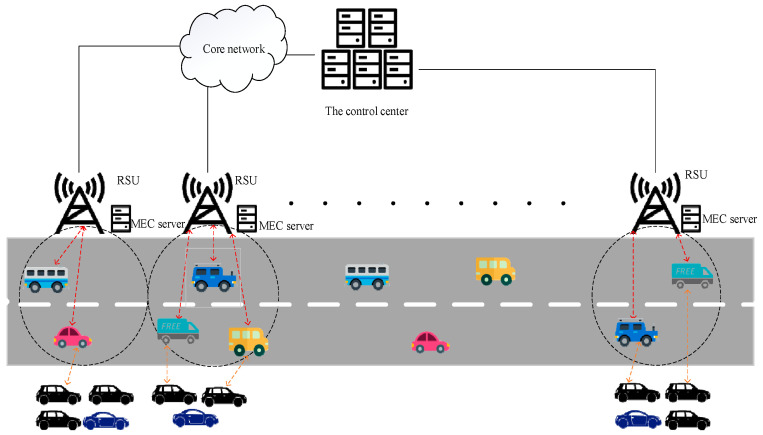
Collaborative mobile edge computing task offloading model for roadside parking.

**Figure 2 sensors-22-04959-f002:**
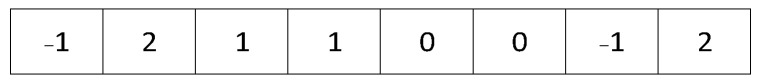
Coding.

**Figure 3 sensors-22-04959-f003:**
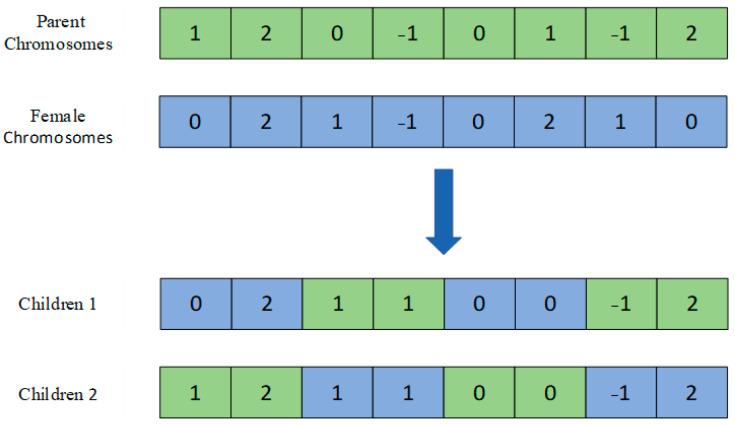
Uniform crossing.

**Figure 4 sensors-22-04959-f004:**
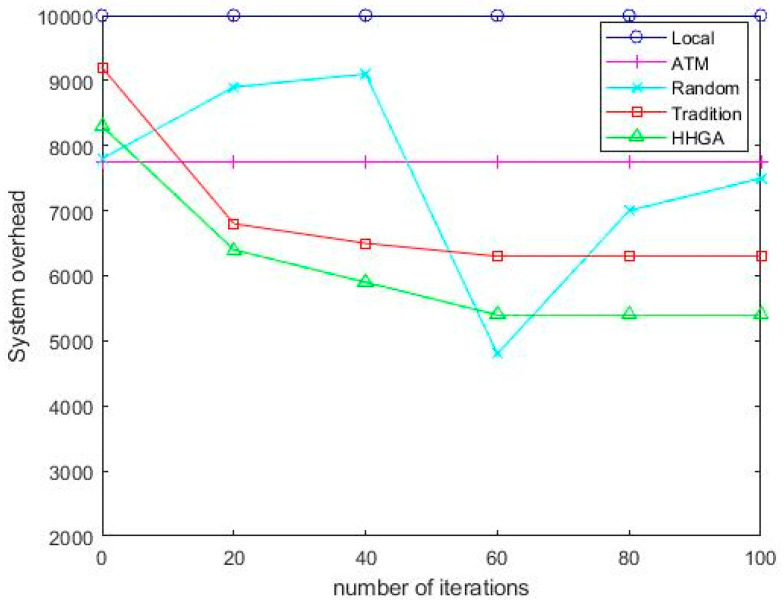
Comparison of offloading schemes.

**Figure 5 sensors-22-04959-f005:**
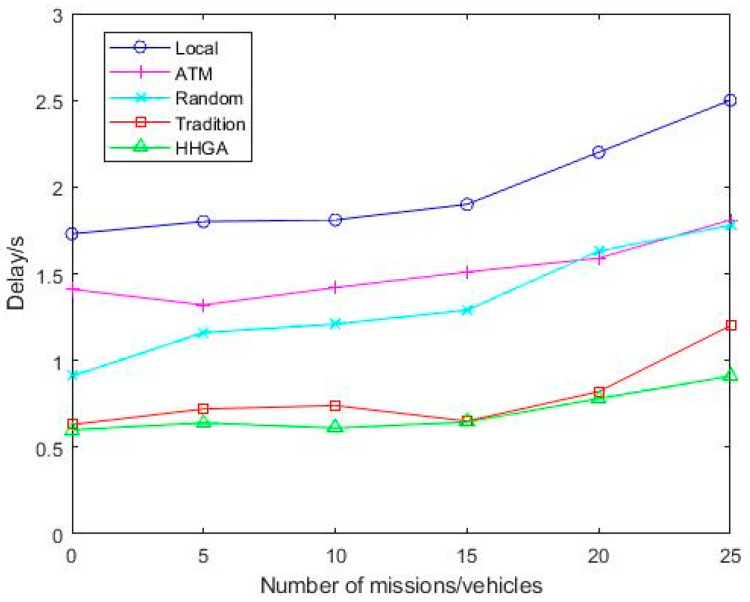
Delay comparison of various algorithms under different number of tasks.

**Figure 6 sensors-22-04959-f006:**
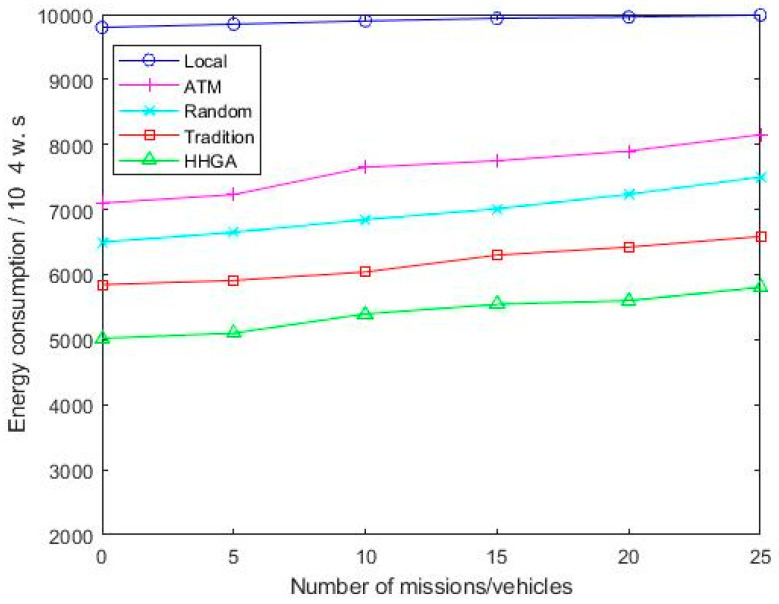
Comparison of energy consumption of various algorithms under different number of tasks.

**Figure 7 sensors-22-04959-f007:**
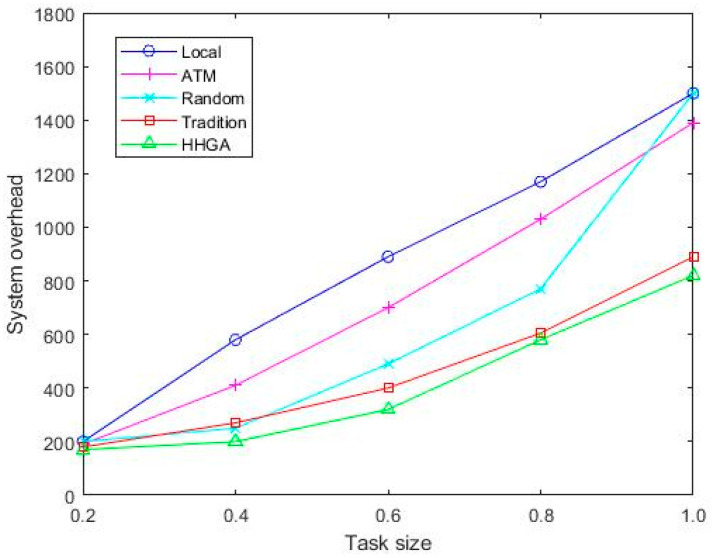
Comparison of system overhead of various algorithms under different task sizes.

**Table 1 sensors-22-04959-t001:** Simulation experiment parameters.

Experimental Parameters	Numerical
The launch rate at which a moving vehicle uploads a task $P_{i}^{up}$	5 W
Computing resources for moving vehicles $f_{i}$	1G cycles/s
Computing resources for MEC $f_{i}^{mec}$	4G cycles/s
Computing resources provided by the cloud server $f_{i}^{cloud}$	10G cycles/s
Curbside parking provides computing resources $f^{side}$	1G cycles/s
Equipment power for moving vehicles/roadside parking $P_{i}$$p_{side}$	8 W
Device power of the MEC server $P_{mecC}$	30 W
Device power of the cloud server $P_{cloud}$	70 W
Populations *M*	60
Maximum number of iterations	100
Crossover rate	0.85
Mutation rate	0.02

## Data Availability

Not applicable.

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
