# Peer review of "Mobile Edge Computing Task Offloading Strategy Based on Parking Cooperation in the Internet of Vehicles"

_sensors, 2022, doi:10.3390/s22134959_

Round 1

Reviewer 1 Report

1.      There is main problem in introduction section so, the authors should be followed these steps:

It is useful to analyze the issues to be considered in the ‘Introduction’ section under 3 headings. Firstly, information should be provided about the general topic of the article in the light of the current literature which paves the way for the disclosure of the objective of the manuscript. Then the specific subject matter, and the issue to be focused on should be dealt with, the problem should be brought forth, and fundamental references related to the topic should be discussed. Finally, our recommendations for solution should be described, in other words our aim should be communicated. When these steps are followed in that order, the reader can track the problem, and its solution from his/her own perspective under the light of current literature.

2.      The authors should be shown more results in the abstract.

3.      Revise the title to make it more meaningful.

4.      Explain the novelty of your work presented in this work.

5.      Algorithm is not clear. The authors should clarify more information in it.

6.      Figures 4, and 5 are very important, the authors should clarify more information in it.

7.      Impact of Task size on the Algorithm performance section is not clear. The authors should clarify more information in it.  

8.      Conclusion was displayed poorly. the author should review this section.

9.      The authors should add more recently references

Reviewer 2 Report

The paper investigates an interesting topic on offloading tasks in a vehicular edge network. The solution proposes using parked vehicles to offload content.

Regarding the presentation of the paper, I have the following comments,

1.     The second paragraph is long, better to be split into at least two paragraphs or three.

2.     The references format is not consistent.

3.     Add a caption/title to the algorithm

4.     In the algorithm, please consider adding the equation used. For instance,

-       Line 4: Calculate Fitness // consider adding the equation used to calculate the fitness.

-       Same thing for all the lines that call mathematical equations from the system model (add which one precisely is used)

5.     Authors are encouraged to add a related works section separately and make an enhanced introduction to motivate their work.

6.     The contributions of the paper should be briefly described in clear sentences. The last contribution “Simulation results show that the proposed scheme can effectively reduce the total system cost during task unloading”, is not a contribution.

From the technical perspective of the paper, I have the following comments,

7.     Authors use the term “unload” and “offload” alternatively, what is the difference? This is confusing the reader.

8.     In the system model, please provide more details about the different strategies (s_i^~), and what are their impact on the system model. i.e., how does these strategies perform offloading

9.     ?

10.  What is the arrival rate of the requests to be offloaded?

11.  Do the authors used any queuing framework to deal with the tasks’ arrival to the MEC infrastructure?

12.  Constraint 1 in the problem formulation needs more clarification, why is it necessary to have a sum less than 1? Is the s_i^~ values binary variables? If yes, add some description to the text (related to the previous point in 8. of this review)

13.  In the problem formulation, constraint 3, how do authors compare the sum of values with a set? Please provide more details. F is a set, and f_i^{mec} are values of this set. Please elaborate.

14.  Regarding the simulations, many details are missing. Please provide a description (in a paragraph) a description of the units used for different values. Please also provide the source of these values.

15.  What does ATM stand for? Please provide the description for all abbreviations used in the paper.

16.  At some level, services migration/handoff between MEC nodes, how would authors deal with such a constraint. Please provide an elaborated discussion regarding services migration, by way of illustration, refer to the following papers,

-       https://doi.org/10.1109/CSCN.2018.8581836

-       https://doi.org/10.1109/LCN52139.2021.9524882

Reviewer 3 Report

The manuscript looks like a technical report rather than a scientific paper. More novelties and theory should be given. At present, the manuscript cannot be accepted for publication. The manuscript should be reorganized and rewritten according to the requirements of a scientific paper. 

Author Response

Thank you for your feedback.

Round 2

Reviewer 2 Report

After checking the authors' response to my comments, I believe that all my comments were addressed, and I suggest they add different answers to my comments accordingly into the final version of the manuscript. 

Author Response

  The author has received your reply, and your previous suggestions have been added into the manuscript. Thanks again.

Reviewer 3 Report

1. Some newly published references should be added in the Section Introduction.

2.The writing should be improved.

3. The sub-titles should be decreased in Section 5. For example, the Sections 5.3 and 5.4 can be merged.
